# Body mass index is associated with hypoglycaemia in children with acute vomiting and dehydration

Simona Tarnokova[1], Lenka Langerova[2], Sabina Szoradova[2], Barbora Pitekova[3,4], Tomas Dallos[2], Juraj Stanik [2,5]*

1 Department of Laboratory Medicine, National Institute for Children´s Diseases, Limbova, Bratislava, Slovakia, 2 Department of Pediatrics, Medical Faculty of the Comenius University and National Institute for Children's Diseases, Limbova, Bratislava, Slovakia, 3 Department of Pediatric Emergency Medicine, Medical Faculty of the Comenius University and National Institute for Children's Diseases, Limbova, Bratislava, Slovakia, 4 Department of Pediatric Urology, Medical Faculty of the Comenius University and National Institute of Children's Diseases, Bratislava, Slovakia, 5 Department of Metabolic Disorders, Institute of Experimental Endocrinology, Biomedical Research Center, Slovak Academy of Sciences, Dubravska Cesta, Bratislava, Slovakia

* juraj.stanik@savba.sk

## Abstract

### Background

Healthy children from 7 months to 7 years are known to be at risk for developing hypoglycaemia during prolonged fasting, particularly during acute illness with decreased oral intake. Our study aimed to identify additional factors associated with hypoglycaemia in children with acute vomiting and dehydration.

### Methods

Our retrospective single-centre study included 560 healthy children and adolescents (aged 29 days to 17.96 years) without known metabolic disorders admitted to hospital with dehydration due to acute illness with vomiting or poor oral intake. Historical and anthropometric parameters were evaluated as potential factors associated with hypoglycaemia.

### Results

A total of 171 (30.5%) participants (aged 0.6–10.7, median 3.8 years) experienced hypoglycaemia (≤3.3 mmol/l). Besides known factors such as a higher degree of dehydration (OR 2.505, 95% CI 1.532–4.095) and complete absence of oral intake (OR 2.185, 95% CI 1.331–3.586), additional factors independently associated with hypoglycaemia included diarrhoea (OR 0.178, 95% CI 0.068–0.468) and lower body mass index (BMI) (OR 0.011, 95% CI 0.000–0.605). Children with hypoglycaemia had a significantly lower BMI (median 14.29 vs 15.46 kg/m², p < 0.001) than children

**Data availability statement:** The dataset contains potentially identifiable clinical information (including exact age and sex) and therefore cannot be fully anonymized to a standard suitable for unrestricted public sharing. The Ethics Committee of the National Institute of Children's Diseases (Bratislava, Slovakia) has restricted public data sharing. De-identified data supporting the findings of this study are available upon reasonable request from the secretary of the Ethics Committee of the National Institute of Children's Diseases, Bratislava, Slovakia (email: detska.klinika@nudch.eu), for researchers who meet the criteria for access to confidential data, subject to Ethics Committee approval and a data-sharing agreement where applicable.

**Funding:** This work was supported by research grant supported by the Scientific Grant Agency of the Ministry of Education, Science, Research and Sport of the Slovak Republic and the Slovak Academy of Sciences VEGA1/0659/22 (awarded to JS), https://www.minedu.sk/vedec-ka-grantova-agentura-msvvam-sr-a-sav-vega/. The funders did not play any role in the study design, data collection and analysis, decision to publish, or preparation of the manuscript.

**Competing interests:** The authors have declared that no competing interests exist.

without hypoglycaemia. Only one child with hypoglycaemia was obese. The highest rate of hypoglycaemia (37.5–51.6%) was observed in the 2–7-year age groups, who also had the lowest median BMI values (13.9–14.8).

## Conclusions

Low BMI and absence of diarrhoea were associated with increased odds of hypoglycaemia. The typical BMI curve for children with physiologically low values at 2–7 years of age may partially explain the high incidence of hypoglycaemia in otherwise healthy children with decreased oral intake at this age.

## Introduction

Hypoglycaemia is common in children [1–3], especially during episodes of vomiting, gastroenteritis or acute infections with decreased oral intake [4–6], and is usually accompanied by increased ketogenesis. In this setting, hypoglycaemia may occur either as idiopathic ketotic hypoglycaemia (IKH) [4,7] or as acute illness-related hypoglycaemia [5,6,8].

Idiopathic ketotic hypoglycaemia is defined by recurrent episodes of ketotic hypoglycaemia following a complete diagnostic evaluation in which alternative causes have been excluded [4]. IKH can be triggered by prolonged fasting or impaired oral intake typically in children between 7 months and 7 years of age [2,4]. The underlying aetiology and factors associated with the development of IKH remain largely unknown [9]. The decreased ability to tolerate prolonged fasting in these children is thought to be due to an inappropriate increase in gluconeogenesis as a result of low levels of gluconeogenetic amino acids, particularly alanine [9,10]. Also, in younger children there is a relatively higher glucose requirement by the brain resulting from the greater brain mass-to-body mass ratio [11]; thus, children with IKH may not be able to satisfy these requirements due to inadequate endogenous glucose production [9]. However, IKH remains a diagnosis of exclusion, and more severe causes of ketotic hypoglycaemia, including hormonal (cortisol and growth hormone deficiency) and metabolic (glycogen storage disease) disorders, have to be ruled out [4,12,13].

By contrast to IKH, hypoglycaemia observed during vomiting or acute infections often represents a first, transient episode, and affected children have typically not yet undergone comprehensive diagnostic work-up [8]. Consequently, studies of hypoglycaemia in acute illness frequently include a heterogeneous population comprising both children who may later meet criteria for IKH and those with acute illness-related hypoglycaemia [1,2]. These two entities therefore substantially overlap in clinical practice.

Early recognition of hypoglycaemia is essential in both idiopathic ketotic hypoglycaemia and illness-related hypoglycaemia. Moreover, hypoglycaemia accompanying gastroenteritis and vomiting has an immediate impact on the management and prognosis of children. Children with hypoglycaemia require acute treatment with glucose, intensive monitoring and often admission to hospital. Appreciation of factors

associated with hypoglycaemia could contribute to early diagnosis. Age, prolonged fasting, or decreased oral intake are commonly reported factors associated with idiopathic ketotic hypoglycaemia [4,7,14]. While the latter are expected contributors, age, in particular the limitation of IKH to children aged 7 months to 7 years [2,4], with a maximum prevalence around 3 years [4], is not clearly explained. Daly et al. [2] suggested that Caucasian origin, male gender, and low body weight may be associated with an increased occurrence of IKH. Illness-related hypoglycaemia shares some of these associations, particularly young age and prolonged fasting, but has also been linked to illness-specific factors, including the duration of vomiting and female sex [5,6,8,15] (Table in S1 Table).

Despite several studies, specific clinical or anthropometric factors particularly associated with illness-related hypo-glycaemia beyond age and fasting-related mechanisms have not been consistently identified. In particular, body mass index (BMI) and BMI standard deviation score (BMI-SDS) have not been systematically evaluated as independent factors associated with hypoglycaemia. We therefore hypothesised that additional factors, potentially related to age-associated physiological changes, may influence the occurrence of hypoglycaemia. This hypothesis was informed by repeated clinical observations of a gracile body habitus and low body weight in children presenting with hypoglycaemia. To verify our hypothesis, we retrospectively analysed the prevalence and factors associated with hypoglycaemia in otherwise healthy children and adolescents hospitalised with dehydration due to vomiting or poor oral intake.

## Methods

### Study participants

A total of 560 children hospitalised at the Department of Paediatrics in the National Institute for Children's Diseases in Bratislava, the largest tertiary paediatric referral centre providing specialized care for children with chronic and rare diseases in Slovakia, from 1 January 2014 to 30 July 2023 were included in the retrospective analysis. First, a search was conducted in the medical data system for children with vomiting and dehydration, and 942 children from age 14 days to 19 years were identified. Subsequently, 346 individuals with chronic diseases (diabetes mellitus, diabetes insipidus, inborn errors of metabolism, idiopathic bowel diseases, multiple congenital anomalies, neuromuscular diseases, Down syndrome), 18 children with normal hydration, and 10 with missing data were excluded.

The remaining children were considered "otherwise healthy" based on the absence of a documented history of chronic disease or previously diagnosed metabolic or hormonal disorders. Owing to the retrospective design, systematic metabolic or hormonal screening was not performed in all participants; therefore, metabolic and hormonal disorders were assumed to be absent unless clinically apparent or previously diagnosed.

As neonates have specific types and factors of hypoglycaemia, an additional 8 children were excluded due to age at admission ≤ 28 days. In the case of repeated hospitalisations, the data from the first documented episode (at the youngest age of the child) with complete data was included in the study. A summary of the study workflow is displayed in Fig 1.

### Historical data

Patient records of the included individuals were searched for baseline data of age, sex, number and duration of vomiting, duration of decreased oral intake (partial or complete), body temperature, and diarrhoea. The severity of dehydration was assessed by a paediatrician according to clinical symptoms (mild dehydration: weight loss <5% in children aged below 2 years and <3% in children older than 2 years of age, no clinical symptoms; moderate dehydration: weight loss <10% in children aged below 2 years and <6% in children older than 2 years of age, decreased skin turgor, thrush on the tongue, oliguria, capillary refill 2–3 s; severe dehydration: weight loss >10% in children aged below 2 years and >7% in children older than 2 years of age, signs of centralisation of the circulation, capillary refill >3 s, or arterial hypotension).

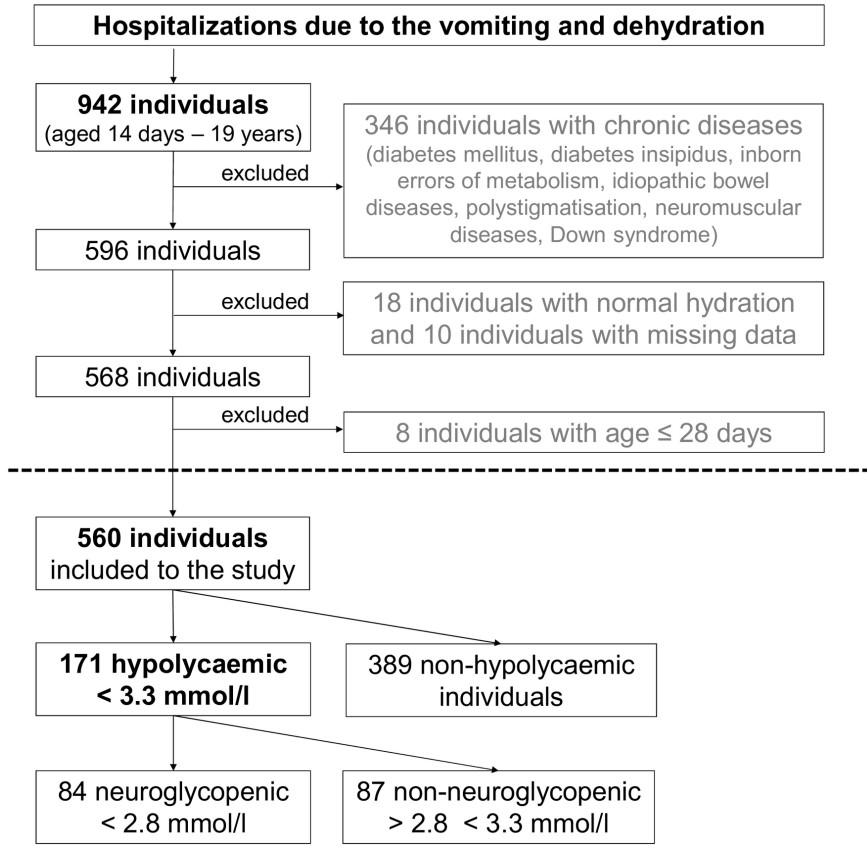

**Fig 1. Study workflow.** Flow chart documenting the inclusion criteria, exclusion criteria, and the number of individuals included in the study.

## Anthropometry

Anthropometric measurements were taken at the time of admission by trained nurses according to standardised protocols. Body mass index (BMI) was calculated as the weight divided by the square of the body height. The standard deviation scores (SDS) for BMI and height were calculated using using age- and sex-specific reference values [16]. BMI-SDS values ≤ −1.88 were evaluated as underweight, values between −1.88 and 1.281 as normal weight, ≥ 1.281 as overweight, and BMI-SDS values ≥ 1.88 as obesity.

## Biochemical analyses

Blood samples were taken within an hour after the child arrived at hospital. The samples were processed and analysed as routine fresh samples by the clinical service laboratory at the National Institute for Children's Diseases in Bratislava. The pH, blood gas analysis, and glucose values were determined in capillary blood samples taken at the time of admission. Serum levels of urea, creatinine, uric acid, sodium, chlorides, and 3-OH butyrate and urine parameters (i.e., the specific gravity of urine, ketones, and pH) were measured using standardised methods. The anion gap (AG) was calculated using the serum concentrations of Na, Cl, $HCO_3^-$, i.e., $AG = sNa - (sCl + sHCO_3^-)$.

## Evaluation of glycaemia

Hypoglycaemia was defined as glycaemia ≤ 3.3 mmol/l based on international recommendations [17]. Additionally, the prevalence of glycaemia ≤ 2.8 mmol/l (the threshold for neuroglycopenia) [17] was determined.

## Statistical analysis

Metric data were checked for normality using the Shapiro–Wilk test. Normally distributed data are expressed as the mean ± standard deviation (SD), and non-normally distributed data (including BMI and serum creatinine) are expressed as the median and interquartile range (25th –75th percentiles). Confidence intervals for percentages in binary data were calculated using the Wilson/Brown method. Differences between the two groups were tested using the t-test for normally distributed data and the Mann-Whitney U test for non-normally distributed metric data; Fisher's test was used for binary data.

All candidate covariates were initially screened using Pearson correlation and univariable logistic regression analyses. To account for multiple testing in the univariable analyses (n = 30), p-values were adjusted using the Bonferroni correction. The Bonferroni-corrected significance threshold was $p < 0.05/30$ (i.e., $p < 0.0017$). In all other analyses, $p < 0.05$ was considered statistically significant.

Multiple logistic regression analyses with hypoglycaemia as the dependent variable were performed in three sequential models: 1. anthropometric co-variables, 2. anamnestic and clinical co-variables, and 3. a combined model including co-variables from steps 1 and 2. Within each model, variables were entered using a forward stepwise selection procedure based on likelihood ratio tests. Non-normally distributed data were log10-transformed prior to regression analyses. All of the covariables were previously tested in Pearson correlation and univariate regression analyses. Independent variables included in the logistic regression models were assessed for collinearity prior to model fitting. Collinearity was evaluated using correlation matrices and variance inflation factors (VIF), and variables showing substantial collinearity were not included simultaneously in the same model. No adjustment for multiple comparisons was applied in the multivariable logistic regression models, as these analyses were hypothesis-driven and based on variables selected from the univariable screening.

**Handling of missing BMI data and sensitivity analyses.** BMI data were missing for 29.8% of participants. The primary analyses were conducted using a complete-case approach. To assess the robustness of the findings to the handling of missing BMI data, two sensitivity analyses were performed. First, the main multivariable logistic regression models were repeated excluding BMI-related variables. Second, missing BMI values were imputed using multiple imputation by chained equations with predictive mean matching, assuming data were missing at random. Imputation was performed in R using the mice package, generating 20 imputed datasets. The imputed datasets were subsequently analysed using the same logistic regression models as in the primary analysis, and results were pooled according to Rubin's rules. For additional exploratory analyses, BMI was categorised relative to the cohort median (BMI below the median, BMI at or above the median, and missing BMI) and entered as a categorical variable in logistic regression models, with BMI at or above the median used as the reference category.

All regression analyses were performed in SPSS, except for the multiple imputation procedure. Statistical analyses were conducted using SPSS version 27 (IBM Corp., Armonk, NY, USA), GraphPad Prism 7 (GraphPad Software, San Diego, CA, USA), and R software.

## Ethics committee

This study was approved by the Ethics Committee of the National Institute of Children's Diseases, Bratislava, Slovakia, and adhered to the tenets outlined in the Declaration of Helsinki. The study was a retrospective analysis of routinely collected clinical data. The requirement for informed consent (written or verbal) from parents or legal guardians of the participating children was waived by the Ethics Committee due to the retrospective design and the use of partially anonymized/de-identified data. Data were accessed for research on December 13, 2023. In response to editorial and reviewer queries, the Ethics Committee issued a formal clarification on December 10, 2025, confirming the waiver of informed consent for this study and approving publication of the results.

 

## Results

### Demographic characteristics

The study included 560 participants aged from 29 days to 17.96 years (median 3.7 years), 299 girls and 261 boys hospitalised with dehydration due to vomiting or poor oral intake. Anthropometric and clinical characteristics of the study cohort are summarised in Table 1, and biochemical and laboratory parameters are presented in Table 2.

### Prevalence of hypoglycaemia and age

Of the 560 participants, 171 (30.5%, CI: 26.9–34.5) experienced hypoglycaemia ≤ 3.3 mmol/l (Fig 1). Children with hypoglycaemia were aged 0.6–10.7 years, median 3.8 years, and more than 95% of them were aged 1–9 years (95.9%, CI: 91.8–98.0). Hypoglycaemia was most prevalent in children aged 3–4 years (51.6%, CI 39.4–63.6), followed by children aged 2 and 3 years (47.9%, CI 36.9–59.2), and children aged 4 and 5 years (47.6%, CI 35.8–59.7). In children aged 2–7 years, the occurrence of hypoglycaemia was > 35%, with an average hypoglycaemia frequency of 47.2% (CI 41.5–53.0) (Fig 2). The lowest prevalence of hypoglycaemia was noted in children aged > 9 years (1.6%, CI: 0.1–8.4) and in children < 1 year of age (6.8%, CI: 3.2–14.1) (Fig 2).

The frequency of hypoglycaemia among girls was 34.1% (CI: 30.0–39.7) and in boys 26.4% (CI 21.5–32.1) (OR 1.441, p = 0.053) (Table 1).

### Severity of hypoglycaemia

Eighty-four children (15.0%, CI: 12.3–18.2) had glycaemia ≤ 2.8 mmol/l, i.e., below the threshold for neuroglycopenia, with similar distribution between age groups as for the ≤ 3.3 threshold (Fig 2). Only two children (aged 1 and 3.5 years, respectively) had hypoglycaemia < 2.0 mmol/l (1.8 and 1.9 mmol/l, respectively). Five children (two boys, three girls, aged 1.4,

**Table 1. Anthropometric and clinical characteristics of children with and without hypoglycaemia.**

| Variable | All participants | Non-hypoglycaemic | Hypoglycaemic | p-value |
|---|---|---|---|---|
| Sex (% female) | 53.4 (560) | 50.6 (389) | 59.6 (171) | 0.054 |
| Age (years)* | 3.70 (1.72–6.13) (560) | 3.54 (1.19–7.05) (389) | 3.80 (2.61–5.27) (171) | 0.543 |
| Height (cm) | 102.37 ± 27.95 (392) | 101.91 ± 31.51 (286) | 103.59 ± 14.58 (106) | 0.473 |
| Height SDS | −0.19 ± 1.02 (393) | −0.23 ± 1.06 (286) | −0.10 ± 0.92 (107) | 0.273 |
| Weight (kg)* | 15.00 (11.10–20.00) (544) | 15.00 (9.60–22.73) (378) | 15.20 (12.98–18.00) (166) | 0.704 |
| Weight SDS* | −0.65 (−1.17–0.00) (544) | −0.61 (−1.17–0.14) (378) | −0.61 (−1.17–0.14) (166) | 0.415 |
| BMI (kg/m²)* | 15.05 (13.92–16.81) (393) | 15.46 (14.17–17.18) (286) | 14.29 (13.47–15.57) (107) | **<0.001** |
| BMI SDS* | −0.68 (−1.21–0.20) (393) | −0.57 (−1.18–0.29) (286) | −0.69 (−1.16—0.16) (107) | **0.021** |
| Fever (% yes) | 59.7 (559) | 60.1 (388) | 59.1 (171) | 0.852 |
| Diarrhoea (% yes) | 20.0 (559) | 25.5 (388) | 7.6 (171) | **<0.001** |
| Degree of dehydration** | 1.47 ± 0.54 (560) | 1.38 ± 0.54 (389) | 1.70 ± 0.49 (171) | **<0.001** |
| Oral intake (% zero intake) | 38.5 (548) | 32.4 (377) | 52.0 (171) | **<0.001** |
| Number of vomiting (decreased intake)* | 3 (0–7) (290) | 2 (0–6) (222) | 6 (3.3–10) (68) | **<0.001** |
| Duration of vomiting (h, decreased intake)* | 24.0 (7.0–48.0) (312) | 24.0 (3.3–48.0) (232) | 43.5 (24.0–55.3) (80) | **<0.001** |
| Number of vomiting (zero intake)* | 7 (4–11) (174) | 7 (4–11) (106) | 6 (4–10) (68) | 0.627 |
| Duration of vomiting (h, zero intake)* | 16.5 (7.0–30.0) (200) | 12.0 (5.3–24.0) (117) | 24.0 (12.0–48.0) (83) | **<0.001** |

Normally distributed data are expressed as the mean ± standard deviation. Non-normally distributed data are presented as the median and interquartile range and are marked with an asterisk (*). Abbreviations: BMI – body mass index; SDS – standard deviation score. Degree of dehydration was classified as mild = 1, moderate = 2, and severe = 3. Values in parentheses indicate the number of participants. P values represent differences between non-hypoglycaemic and hypoglycaemic children. Univariable p-values were adjusted for multiple testing using the Bonferroni correction (30 tests); p-values < 0.05/30 were considered statistically significant and are shown in bold.

**Table 2. Biochemical and laboratory characteristics of children with and without hypoglycaemia.**

| Variable | All participants | Non-hypoglycaemic | Hypoglycaemic | p-value |
|---|---|---|---|---|
| **Glycaemia (mmol/l)** | 4.30±1.38 (560) | 4.95±1.14 (389) | 2.83±0.33 (171) | **<0.001** |
| **Serum β-hydroxybutyrate (mmol/l)** | 4.22±2.34 (45) | 3.51±2.43 (26) | 5.20±1.85 (19) | 0.015 |
| **Serum urea (mmol/l)** | 4.92±2.05 (501) | 4.46±2.09 (346) | 5.94±1.52 (155) | **<0.001** |
| **Serum creatinine (µmol/l)*** | 35.0 (28.0–42.0) (534) | 33.0 (26.0–44.0) (367) | 37.0 (32.0–42.0) (167) | 0.002 |
| **Serum uric acid (µmol/l)** | 416.33±191.11 (489) | 353.38±174.67 (336) | 554.57±148.41 (153) | **<0.001** |
| **Serum sodium (mmol/l)** | 137.56±3.64 (559) | 137.96±3.73 (388) | 136.63±3.25 (171) | **<0.001** |
| **Serum chloride (mmol/l)** | 103.91±4.66 (557) | 104.35±4.84 (386) | 102.92±4.08 (171) | **0.001** |
| **Capillary pH** | 7.39±0.08 (527) | 7.41±0.07 (356) | 7.34±0.06 (171) | **<0.001** |
| **Capillary $pCO_2$ (kPa)** | 3.44±0.75 (527) | 3.63±0.78 (356) | 3.05±0.51 (171) | **<0.001** |
| **Base excess (mmol/l)** | −7.56±5.01 (527) | −5.65±4.56 (356) | −11.54±3.27 (171) | **<0.001** |
| **$HCO_3^-$ (mmol/l)** | 15.85±4.61 (527) | 17.61±4.23 (356) | 12.18±2.90 (171) | **<0.001** |
| **Anion gap (mmol/l)** | 17.72±5.90 (524) | 15.87±5.40 (353) | 21.54±4.98 (171) | **<0.001** |
| **Urine ketones (% positive)** | 78.6 (387) | 68.6 (261) | 99.2 (126) | **<0.001** |
| **Urine specific gravity** | 1021.91±10.19 (387) | 1019.31±10.65 (261) | 1027.29±6.43 (126) | **<0.001** |
| **Urine pH** | 5.67±0.71 (388) | 5.77±0.74 (261) | 5.45±0.59 (127) | **<0.001** |

Normally distributed data are expressed as the mean±standard deviation. Non-normally distributed data are presented as the median and interquartile range and are marked with an asterisk (*). Values in parentheses indicate the number of participants. P values represent differences between non-hypoglycaemic and hypoglycaemic children. Univariable p-values were adjusted for multiple testing using the Bonferroni correction (30 tests); p-values < 0.05/30 were considered statistically significant and are shown in bold.

1.9, 2.0, 3.5 and 3.7 years) had impaired consciousness during hypoglycaemia (2.7, 2.8, 2.9, 3.1, and 3.2 mmol/l, respectively) and two of them had seizures.

## Type of hypoglycaemia

The majority of the hypoglycaemic individuals (125 out of 126; 99.2%, CI 95.6–100.0) with available urine samples at the time of hospital admission had positive ketones in the urine and thus ketotic hypoglycaemia. Positive urine ketones were less frequent in the non-hypoglycaemic group (68.6%, CI: 62.7–73.9) (p<0.001, OR 57.3 CI: 10.5–581.9) (Table 2). In 43 children, plasma beta-hydroxybutyrate was available at the time of admission, with all results being above the normal range of 0.03–0.3 mmol/l (Table 2); 18 (42%) of them had hypoglycaemia ≤ 3.3 mmol/l. Seven children (four girls, and three boys, aged 2.3–8.4 years) were repeatedly hospitalised for dehydration, two of them also with hypoglycaemia. No other defined causes of hypoglycaemia were identified in children with hypoglycaemia.

## Hypoglycaemia and anthropometric parameters

No differences in weight and weight SDS were found between hypoglycaemic and non-hypoglycaemic participants (Table 1). However, all children with hypoglycaemia ≤ 3.3 mmol/l weighed ≤ 30 kg (Fig 3) (and all children with more severe hypoglycaemia of ≤ 2.8 mm/l weighed ≤ 21.5 kg). There were no significant differences in height and height-SDS between groups.

Children with hypoglycaemia had a lower BMI (median 14.3, interquartile range 13.5–15.6 kg/m², p<0.001) (Fig 3). The highest frequency of hypoglycaemia (37.5–51.6%) was observed in the age groups of 2–7 years, which also had low median BMI values (13.9–14.8). Moreover, the frequency of hypoglycaemia was negatively correlated with median BMI in each age group (r=−0.711, p=0.021) (Fig 4).

The majority of participants with available BMI had BMI-SDS values in the normal range, i.e., between the 3rd and 97th percentile (99/107); seven were < 3rd percentile, and one had BMI-SDS > 97th percentile. Children with

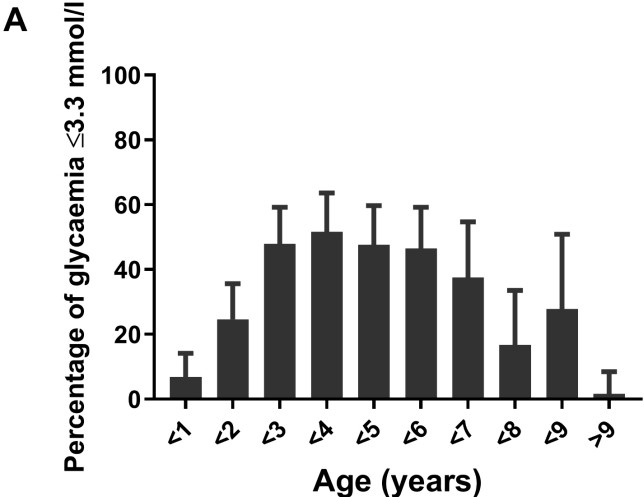

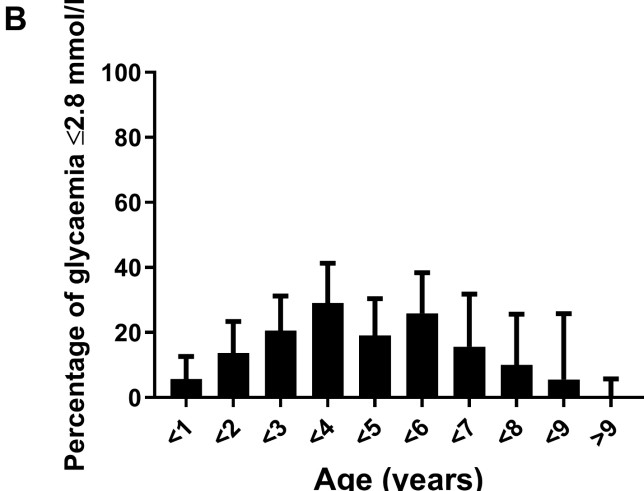

**Fig 2. Frequency of hypoglycaemia in different age groups. A.** Frequency of hypoglycaemia ≤3.3. mmol/l; **B.** Frequency of glycaemia ≤ 2.8 mmol/l; Confidence intervals (CI) for percentages were calculated using the Wilson/Brown method.

hypoglycaemia had lower BMI-SDS (p = 0.021) compared to non-hypoglycaemic participants (Table 1, Fig 3). Only one child with obesity (BMI-SDS 1.9) was recorded in the hypoglycaemic group (1.2%, CI: 0.1–6.4), whereas 17 obese children (5.9%, CI: 3.7–9.3) were in the non-hypoglycaemic group; however the differences were not significant (p = 0.087) (Fig 3).

**Missing BMI data.** In exploratory logistic regression analyses, missing BMI values were significantly associated with hypoglycaemia. Compared with children with BMI at or above the median, children with missing BMI had higher odds of hypoglycaemia (OR 2.67, 95% CI 1.66–4.29, p < 0.001). Similarly, children with BMI below the median also had increased odds of hypoglycaemia compared with those with BMI at or above the median (OR 2.37, 95% CI 1.49–3.76, p < 0.001). These findings are consistent with the observed prevalence data and indicate that missing BMI values cluster in children with a risk profile similar to that of lower BMI.

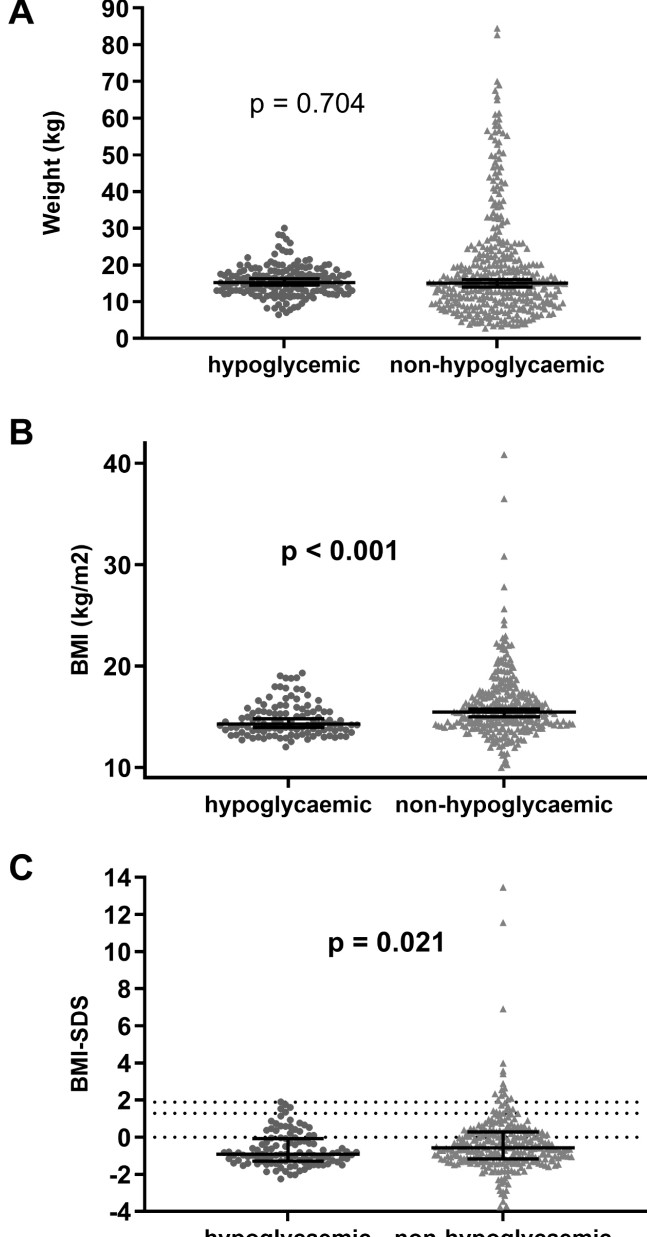

**Fig 3. Selected anthropometric parameters in hypoglycaemic and non-hypoglycaemic groups.** Selected anthropometric parameters were A. weight, **B.** BMI, and **C.** BMI-SDS. The black line with error bars represent the median with interquartile ranges. Differences between the two groups were calculated using the Mann-Whitney U test. The dotted lines in figure C represent BMI-SDS cut-offs for overweight (the grey line at a BMI-SDS value of 1.281) and for obesity (the black line at a BMI-SDS value of 1.88).

## Association of hypoglycaemia with dehydration and vomiting

Individuals with hypoglycaemia had a higher degree of dehydration (used as ordinary numeric variable: mild = 1, moderate = 2, severe = 3) ($p < 0.001$) and less frequently diarrhoea ($p < 0.001$) compared with the non-hypoglycaemic group. The presence of fever did not significantly differ between the hypoglycaemic and non-hypoglycaemic groups (Table 1).

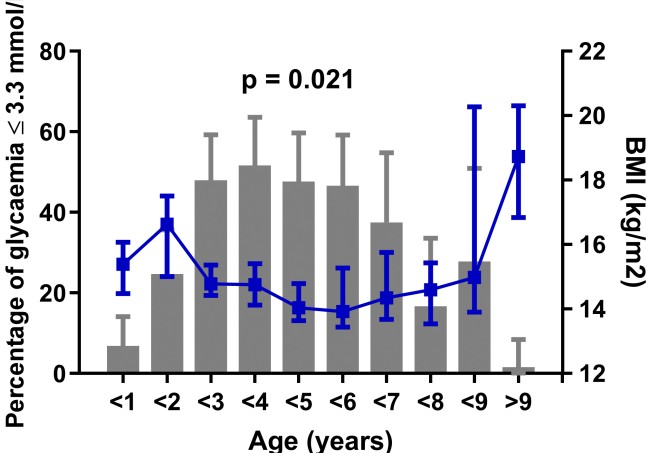

**Fig 4. Percentage of children with hypoglycaemia and BMI in different age groups.** Confidence intervals (CI) for percentages were calculated using the Wilson/Brown method. BMI is displayed as the median with interquartile ranges. Association of percentage of hypoglycaemia with BMI was calculated in Pearson correlation.

For further analyses of vomiting and starvation, participants were divided into group 1 (decreased oral intake), and group 2 (no oral intake). In group 1, significant differences in the number of vomiting episodes (p<0.001) and duration of vomiting were found between hypoglycaemic and non-hypoglycaemic children (p<0.001) (Table 1). In group 2, hypoglycaemic children had longer duration of vomiting compared with the non-hypoglycaemic group (p<0.001) (Table 1).

## Association of hypoglycaemia with biochemical parameters

Hypoglycaemia was associated with more severe metabolic acidosis: hypoglycaemic individuals had lower pH, pCO2, base excess and $HCO_3^-$ capillary blood levels compared with the non-hypoglycaemic group (Table 2). Individuals with hypoglycaemia also had a higher anion gap, lower serum concentrations of sodium, potassium and chloride, and also lower urine pH and higher urine specific gravity compared with the non-hypoglycaemic group (p<0.001) (Table 2).

In the serum measurements the hypoglycaemic group had higher concentrations of urea, uric acid, and creatinine (p<0.001), and also higher concentrations of beta hydroxybutyrate serum levels (p=0.015) (Table 2).

## Factors associated with hypoglycaemia in children with dehydration

We divided the variables into two groups, i.e., 1. age and anthropometric data, and 2. historical factors, such as dehydration, fasting and vomiting. Logistic regression with hypoglycaemia as the dependent variable was performed first in each group separately and then in both groups together. Laboratory factors were not included in the analyses, as they are usually examined in parallel with glycaemia.

In the forward stepwise logistic regression with sex and anthropometric data (BMI, BMI-SDS), a significant association with hypoglycaemia was found for low BMI and low BMI-SDS values (p<0.001) (Table 3, Model 1).

In the forward stepwise logistic regression with hypoglycaemia as the dependent variable and with degree of dehydration (used as numeric ordinal variable: mild=1, moderate=2, severe=3), diarrhoea, and oral intake – zero or decreased (both used as binary variables) used as factors, a significant association was found for a higher degree of dehydration (p<0.001), presence of diarrhoea (p<0.001), and zero oral intake (p=0.006) (Table 3, Model 2).

By combining the factors of hypoglycaemia (BMI, BMI-SDS, and degree of dehydration, diarrhoea, and oral intake) in the logistic regression we found a significant association with all of the tested variables (Table 3, Model 3).

**Table 3. Regression analyses of hypoglycaemia and selected co-variables.**

| Model | Model Summary | Dependent | Independent | ΔR2 | OR | CI of OR (95%) | p value |
|---|---|---|---|---|---|---|---|
| 1 | R2 = 0.062, p < 0.001, n = 393 | hypoglycaemia (≤3.3 mmol/l) | BMI (log10 transformed) | 0.062 | 0.001 | 0.000–0.027 | **<0.001** |
| 2 | R2 = 0.167, p < 0.001, n = 547 | hypoglycaemia (≤3.3 mmol/l) | degree of dehydration | 0.101 | 2.806 | 1.934–4.071 | **<0.001** |
| | | | diarrhoea | 0.049 | 0.297 | 0.158–0.560 | **<0.001** |
| | | | zero oral intake | 0.017 | 1.734 | 1.173–2.562 | **0.006** |
| 3 | R2 = 0.248, p < 0.001, n = 382 | hypoglycaemia (≤3.3 mmol/l) | degree of dehydration | 0.119 | 2.505 | 1.532–4.095 | **<0.001** |
| | | | diarrhoea | 0.081 | 0.178 | 0.068–0.468 | **0.001** |
| | | | zero oral intake | 0.031 | 2.185 | 1.331–3.586 | **0.002** |
| | | | BMI (log10 transformed) | 0.017 | 0.011 | 0.000–0.605 | **0.027** |

Analysed in forward stepwise multiple logistic regression analyses. Dependent variable: hypoglycaemia (≤3.3 mmol/l). Covariates: sex, logBMI for Model 1; degree of dehydration, diarrhoea, and zero oral intake (zero = 1, decreased = 0) for Model 2; and logBMI, degree of dehydration, diarrhoea, and zero oral intake for Model 3. Abbreviations: BMI – body mass index.

**Sensitivity analyses.** Results of the sensitivity analyses were consistent with those of the primary complete-case analysis. In the model excluding BMI, the associations between the remaining covariates and hypoglycaemia were unchanged. In the multiple imputation analysis, the association between BMI and hypoglycaemia remained statistically significant, with a comparable effect size and increased statistical precision compared with the complete-case analysis (complete-case analysis: OR 0.011, 95% CI 0.000–0.605, p = 0.007; multiple imputation analysis: OR 0.067, 95% CI 0.015–0.391, p = 0.002). Estimates for the remaining covariates were unchanged. Results of the multiple imputation sensitivity analysis are presented in the S2 Table.

## Discussion

In our study, illness-related hypoglycaemia was found in 30.5% of otherwise healthy children admitted to hospital with vomiting and dehydration. The occurrence of hypoglycaemia was associated with low BMI, a higher degree of dehydration, absence of diarrhoea, and zero oral intake as independent factors in the multiple logistic regression analyses.

### Previously reported factors associated with hypoglycaemia

Ketotic hypoglycaemia occurs during reduced oral intake even in otherwise healthy children [2,4,7]. It most commonly has the character of idiopathic ketotic hypoglycaemia or illness-related hypoglycaemia. In the present study, hypoglycaemia predominantly had the character of illness-related hypoglycaemia, as in most cases it represented the first documented episode of hypoglycaemia. At the time of hospitalisation, the majority of children had not undergone a complete diagnostic evaluation required to confirm or exclude idiopathic ketotic hypoglycaemia. Nevertheless, in a subset of participants, idiopathic ketotic hypoglycaemia was diagnosed later during subsequent hospitalisations or outpatient follow-up.

Hypoglycaemia in both IKH and illness-related hypoglycaemia typically occurs during prolonged fasting, or during vomiting or refusal of oral intake [1,4,15]. In our study, hypoglycaemia was significantly associated with zero oral intake. Duration of vomiting has been previously described as a factor associated with development of illness-related hypoglycaemia [5,6]. In line with previous reports, duration of vomiting was also associated with hypoglycaemia in our cohort. Moreover, the increasing number of vomiting episodes was also associated with hypoglycaemia.

Age is a known factor associated with idiopathic ketotic hypoglycaemia [4,7] as well as with illness-related hypoglycaemia [5]. In our study, the highest incidence of illness-related hypoglycaemia was also observed in the group of children aged between 2 and 7 years, whereas hypoglycaemia was uncommon in participants younger than 1 year or older than 9 years. However, age itself was not identified as an independent factor in multivariable analyses.

Previous work has presented a higher incidence of idiopathic ketotic hypoglycaemia in boys [2,9,10]. Most of these studies were based on relatively small cohorts, limiting the interpretation of sex-related differences. Brown et al. in 2015 [12] summarised previously published patients and found 48 girls and 97 boys with idiopathic ketotic hypoglycaemia in the literature. Using the Chi-Squared test for Goodness of Fit with an expected incidence of IKH in girls and boys of 1:1, the observed incidence of IKH was significantly higher in boys (p<0.001). However, the published papers included in the analysis described children with confirmed IKH who had undergone the entire diagnostic process for IKH. However, this does not automatically imply a higher prevalence in boys in an unselected population of otherwise healthy children with illness-related dehydration. This may explain why we did not find a higher prevalence of hypoglycaemia in boys in our comparably large cohort. Notably, in contrast to studies of IKH, Reid et al. in 2003 [8] identified female gender as an independent factor associated with illness-related hypoglycaemia.

The association between the degree of dehydration and illness-related hypoglycaemia is of particular interest. The severity of dehydration has been considered as a potential factor associated with hypoglycaemia in several previous studies. Bennish et al. [15] and Reid et al. [8] compared the occurrence of mild versus moderate and severe dehydration in children with and without hypoglycaemia. Bennish et al. reported a higher proportion of moderate to severe dehydration among children with hypoglycaemia (37%) compared with those without hypoglycaemia (24%); however, this difference was not statistically significant (p=0.301) [15]. Similarly, Reid et al. did not observe significant differences in the prevalence of more severe dehydration between children with hypoglycaemia (59%) and those without hypoglycaemia (61%) [8]. In contrast, in our study, the degree of dehydration was significantly associated with the occurrence of hypoglycaemia in both univariable and multivariable analyses. This difference may be partly explained by the larger sample size of our cohort and the relatively low proportion of children with diarrhoea, which may have resulted in longer periods of reduced oral intake and fasting. From a pathophysiological perspective, more severe dehydration may serve as a marker of greater metabolic stress and prolonged fasting, conditions that are known to increase susceptibility to hypoglycaemia during acute illness. In addition, assessment of dehydration severity is inherently subjective to some extent, which may also contribute to variability between studies.

Another factor previously reported to be associated with illness-associated hypoglycaemia, and confirmed in our analyses, was metabolic acidosis, likely reflecting increased ketogenesis and metabolic stress during prolonged fasting and acute illness [3].

### Known anthropometric factors associated with hypoglycaemia

Several previous studies have mentioned the association of anthropometric parameters with the occurrence of hypoglycaemia. Children aged<1 year had a low incidence of hypoglycaemia and also low weight; in children aged 2–7 years, both the prevalence of hypoglycaemia and weight increase, but the frequency of hypoglycaemia subsequently decreases as weight increases. Thus, weight alone may not appropriately reflect the capacity of the organism to compensate for hypoglycaemia. Colle et al. showed that children with idiopathic ketotic hypoglycaemia were in the less than 50th percentile for height and weight [7]. Habbick et al. also presented similar results in their study, showing that 19 of 20 reported children with IKH had weight<50th percentile [14]. In contrast, studies focusing on illness-related hypoglycaemia have generally not evaluated anthropometric parameters. The only exception is the study by Bennish et al., which compared weight-for-age and weight-for-height between children with and without hypoglycaemia, but found no significant differences between the groups [15]. Similarly, in our study, we did not find an association between hypoglycaemia and body weight or weight-SDS.

In contrast to body weight, BMI-SDS has been evaluated only rarely as a potential factor associated with hypoglycaemia, and exclusively in the context of idiopathic ketotic hypoglycaemia. In the study by Kaplowitz et al. [4] children with idiopathic ketotic hypoglycaemia had a mean BMI percentile of 46.5, and 5 out of 56 participants had BMI<5th percentile. However, the association of BMI percentile with hypoglycaemia was not calculated, as no controls without hypoglycaemia

were included to the study. Similarly, in our study, the majority of participants with hypoglycaemia had BMI-SDS values in the normal range, i.e., between the 3rd and 97th percentile (99/107), while seven were < 3rd percentile, and one had BMI-SDS > 97th percentile. Moreover, only one child with obesity (BMI-SDS 1.9) was recorded in the hypoglycaemic group (1.2%), whereas 17 children (5.9%) in the non-hypoglycaemic group were obese, although this difference was not significant (p = 0.087). However, BMI-SDS was not identified as an independent factor for illness-related hypoglycaemia in our study, as it did not enter the multivariable models due to Bonferroni correction in univariable analyses and its collinearity with absolute BMI. The reasons may be similar to those for age: previous studies mostly did not have a control group of children without hypoglycaemia, and also the association of weight and hypoglycaemia is not linear. Thus, using a relative measure of body proportions may be more appropriate.

## Body mass index as a novel factor associated with hypoglycaemia

The association of hypoglycaemia – neither idiopathic ketotic hypoglycaemia nor illness-related hypoglycaemia – with BMI has not been previously described. One possible reason why BMI may not have been explored as a factor associated with hypoglycaemia in previous studies may be that absolute BMI is rarely used in paediatric studies, as age- and gender-adjusted BMI z-scores (BMI-SDS) are preferred. For example, Kaplowitz et al. [4] reported BMI percentiles in children with idiopathic ketotic hypoglycaemia, but no comparison with non-hypoglycaemic controls was performed, and no multivariable analyses were used to assess the independent contribution of BMI. Similarly, earlier cohort studies focusing on idiopathic ketotic hypoglycaemia described lower weight or height percentiles [7,14], but did not evaluate absolute BMI or disentangle its effect from age-related physiological changes. In contrast, our study included a control group of children with dehydration and vomiting without hypoglycaemia and applied multivariable modelling, allowing us to identify BMI as an independent factor associated with hypoglycaemia.

BMI was independently associated with hypoglycaemia in children hospitalised with dehydration and vomiting, and this association was stronger than that observed for BMI-SDS, which is routinely used to assess nutritional status in children. This may suggest that total body mass relative to height plays a role in the metabolic capacity to maintain euglycaemia during acute illness, independent of age-related normative adjustments. We could speculate that lower BMI may be usually associated with reduced muscle mass, fat stores, and potentially a smaller liver, which are the key organs involved in gluconeogenesis [18]. Although metabolic efficiency cannot be inferred solely from organ size, a lower total body mass may limit metabolic reserves during fasting or acute illness. However, further studies are needed to clarify the exact relationship between BMI and these metabolic parameters in children with hypoglycaemia. This interpretation is supported by the fact that BMI is physiologically lowest at 2–9 years of age [16], which overlaps with the age of highest prevalence of hypoglycaemia. BMI may therefore reflect a physiologically reduced metabolic reserve of the organism to respond to hypoglycaemia-promoting processes during this developmental period. In contrast, BMI-SDS accounts for this physiological decline and may therefore be less sensitive to detect absolute differences in metabolic capacity.

The observation that children with missing BMI values had similarly increased odds of hypoglycaemia as those with lower BMI suggests that incomplete anthropometric documentation may identify younger or clinically more vulnerable patients. From a clinical perspective, missing BMI data should therefore not be interpreted as indicating low risk, but rather as a prompt for careful glucose assessment.

## Illness-related hypoglycaemia and diarrhoea

The association between diarrhoea and illness-related hypoglycaemia is not entirely novel. Bennish et al. [15] reported that hypoglycaemia was associated with a shorter duration of diarrhoea in children with acute gastroenteritis. Similarly, Reid et al. [8] showed that hypoglycaemia occurred more frequently in children in whom vomiting predominated over diarrhoea. However, in these studies, the presence or absence of diarrhoea was not analysed as an independent factor. In our study, the presence of diarrhoea was independently associated with lower odds of hypoglycaemia. This finding extends previous observations by suggesting that

diarrhoea may not merely coexist with hypoglycaemia, but may be inversely associated with its occurrence in the clinical context of vomiting and dehydration. This association can be explained by the typical temporal pattern of acute gastroenteritis, where vomiting and refusal of oral intake often precede diarrhoea. The presence of diarrhoea may therefore indicate a later phase of illness with partial recovery of oral intake and reduced fasting-related metabolic stress. Differences in energy intake, fluid and electrolyte balance, and metabolic demands during different phases of illness may further contribute to this association.

## Strengths and Limitations

A key strength of our study lies in the characteristics of the study cohort. Compared with cohorts focusing on idiopathic ketotic hypoglycaemia [4,7,14], which typically include small numbers of carefully phenotyped patients after extensive diagnostic work-up, our cohort is substantially larger, allowing for more robust statistical analyses and the identification of additional factors associated with hypoglycaemia. Conversely, in comparison with studies of illness-related hypoglycaemia conducted primarily in emergency department settings [6,8], our cohort benefits from more detailed clinical and anthropometric phenotyping, including standardized assessment of dehydration severity, vomiting characteristics, and laboratory parameters. This intermediate position between highly selected IKH cohorts and large but often sparsely characterised emergency department populations provides a complementary perspective and allows a more nuanced evaluation of factors associated with hypoglycaemia in otherwise healthy children presenting with acute illness.

However, the retrospective nature of the study is associated with several limitations and potential sources of bias. First, selection bias related to hospital admission and referral cannot be excluded. As a tertiary referral centre, our institution is more likely to admit children with more severe dehydration, prolonged vomiting, or clinical deterioration, which may have inflated the observed prevalence of hypoglycaemia compared with unselected populations presenting to primary care or outpatient settings. Therefore, the reported prevalence should be interpreted in the context of hospitalised children requiring inpatient care rather than the general paediatric population with acute illness.

Second, anthropometric and clinical data were extracted from medical records and therefore depended on the completeness and accuracy of routine clinical documentation. In particular, BMI data were missing for a proportion of participants, which may have introduced selection bias. To address this limitation, we performed sensitivity analyses using both a complete-case approach and multiple imputation of missing BMI values. The consistency of results across these analyses supports the robustness of the observed associations and reduces the likelihood that missing BMI data substantially biased the main findings.

Measurements of serum beta hydroxybutyrate concentrations were not available in a significant proportion of included children, limiting the ability to biochemically characterise ketosis in all cases. Furthermore, as this was an observational study in routine clinical practice, most children with hypoglycaemia did not undergo a comprehensive differential diagnostic workup to exclude rare metabolic or endocrine disorders. Consequently, the presence of undiagnosed underlying conditions cannot be entirely ruled out.

An additional limitation of this study is the partial overlap between idiopathic ketotic hypoglycaemia and hypoglycaemia occurring during acute illness. Due to the retrospective design, it was not possible to reliably distinguish classical IKH from hypoglycaemia secondary to acute illness in all cases. However, this reflects real-world clinical practice, where extensive investigations are usually reserved for recurrent or severe episodes. Accordingly, our findings should be interpreted as describing factors associated with illness-related hypoglycaemia in otherwise healthy children, rather than exclusively classical IKH.

Diagnosis of hypoglycaemia based on Whipple's triad (symptoms and/or signs consistent with hypoglycaemia, a documented low plasma glucose concentration, and relief of signs/symptoms when plasma glucose concentration is restored to normal) [17] was not generally applicable, as most of the participants were infants unable to reliably symptoms. Moreover, dehydration, ketosis, metabolic acidosis and underlying infection could have symptoms hardly distinguishable from those in hypoglycaemia (e.g., apathy, lethargy).

Finally, as with all retrospective observational studies, residual confounding by unmeasured factors cannot be excluded, and the identified associations should not be interpreted as causal.

## Implications for clinical practice

As this was a retrospective observational study, our findings demonstrate associations rather than causal relationships. Low BMI and absence of diarrhoea were associated with increased odds of hypoglycaemia. These factors, readily identifiable at presentation, may support early consideration of blood glucose screening in children presenting with vomiting and dehydration. However, they should be considered supportive clinical markers rather than validated predictors, as causal relationships cannot be inferred from the present study design. Therefore, prospective validation is required before these factors can be used as predictive tools or screening criteria in clinical practice.

## Conclusions

Hypoglycaemia during vomiting or reduced oral intake is common in children and may occur in more than 30% of those requiring medical attention for dehydration. Low BMI and absence of diarrhoea were independently associated with hypoglycaemia in this cohort. Recognition of such factors may contribute to earlier identification of children at increased risk; however, prospective studies are needed to validate their predictive value and clinical applicability.

## Supporting information

**S1 Table. Published original studies reporting clinical factors associated with hypoglycaemia in children: idiopathic ketotic hypoglycaemia versus hypoglycaemia during acute illness.** Only original studies including children beyond the neonatal period were included. Studies focusing on hypoglycaemia due to congenital hyperinsulinism, inborn errors of metabolism, fatty acid oxidation defects, glycogen storage diseases, medication-induced hypoglycaemia, or neonatal hypoglycaemia were excluded. Abbreviations: IKH, idiopathic ketotic hypoglycaemia; ED, emergency department; BMI, body mass index; y, years.
(DOCX)

**S2 Table. Multivariable logistic regression analysis of factors associated with hypoglycaemia using multiple imputation for missing BMI data.** Multivariable logistic regression analyses were performed with hypoglycaemia (≤3.3 mmol/l) as the dependent variable. Covariates included sex, log-transformed BMI (imputed), degree of dehydration, diarrhoea, and oral intake (zero = 1, decreased = 0). Missing BMI values were imputed using multiple imputation by chained equations with predictive mean matching. The imputed datasets were analysed using the same multivariable logistic regression model as in the primary analysis, and regression estimates were pooled across imputations according to Rubin's rules. Abbreviations: BMI, body mass index.
(DOCX)

## Acknowledgments

We are grateful to Daniela Gasperikova, DSc. and Martina Skopkova, Ph.D. for contribution to the Discussion, and also to all the doctors and nurses from the Department of Paediatrics in Bratislava who cared for the children enrolled in the study.

## Author contributions

**Conceptualization:** Tomas Dallos, Juraj Stanik.

**Data curation:** Simona Tarnokova, Lenka Langerova, Sabina Szoradova, Barbora Pitekova, Juraj Stanik.

**Formal analysis:** Simona Tarnokova, Juraj Stanik.

**Funding acquisition:** Juraj Stanik.

**Investigation:** Juraj Stanik.

**Methodology:** Juraj Stanik.

**Project administration:** Juraj Stanik.

**Resources:** Juraj Stanik.

**Supervision:** Tomas Dallos, Juraj Stanik.

**Validation:** Tomas Dallos, Juraj Stanik.

**Visualization:** Juraj Stanik.

**Writing – original draft:** Simona Tarnokova, Juraj Stanik.

**Writing – review & editing:** Tomas Dallos, Juraj Stanik.

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
