## [Decision Letter · Decision Letter 0]

9 Dec 2025

Dear Dr. Stanik,

Thank you for submitting your manuscript to PLOS ONE. After careful consideration, we feel that it has merit but does not fully meet PLOS ONE’s publication criteria as it currently stands. Therefore, we invite you to submit a revised version of the manuscript that addresses the points raised during the review process.

We look forward to receiving your revised manuscript.

Kind regards,

Tacilta Nhampossa

Academic Editor

PLOS One

Journal Requirements:

5. Please provide a complete Data Availability Statement in the submission form, ensuring you include all necessary access information or a reason for why you are unable to make your data freely accessible. If your research concerns only data provided within your submission, please write "All data are in the manuscript and/or supporting information files" as your Data Availability Statement.

Additional Editor Comments :

Summary Report:

-     This retrospective study investigates risk factors for hypoglycemia in otherwise healthy children admitted with acute vomiting and dehydration. Among 560 patients aged 29 days to 17.9 years, 30.5% developed hypoglycemia (≤3.3 mmol/L), predominantly in the 2–7 year age range. Besides known contributors such as decreased oral intake and dehydration, the study identifies low BMI, low BMI-SDS, and diarrheas' as independent predictors. The findings offer a plausible mechanistic link between physiologically low BMI in early childhood and heightened susceptibility to hypoglycemia. No description of the study hospital

Comments

1.     The manuscript is generally clear, but occasional grammatical issues should be corrected (e.g., “IKT” should be “IKH”; “polystigmatisation” may be unclear in English and should be defined or replaced). Some sentences are overly long; breaking them into shorter units would improve readability.

2.     Ensure consistent use of terms (e.g., “hypoglycaemia ≤ 3.3 mmol/L” vs. “hypoglycaemia < 3.3 mmol/L”).

3.     Define all abbreviations at first use (e.g., AG for anion gap, SDS, IKH).

4.     Originality and Significance:

-     The link between BMI and hypoglycemia is interesting and clinically relevant, especially since the 2–7 year age window coincides with physiologic BMI nadir. This provides a valuable explanatory model for age-related vulnerability. However, the claim of novelty could be better contextualized with more detailed comparison to prior studies (e.g., Daly et al., other hypoglycaemia risk factor cohorts). Consider expanding the Introduction/Discussion to highlight how your findings go beyond known associations.

5.     There is a poor description of the   National Institute for  Children’s Diseases in Bratislava

6.     Study Design and Limitations

-     The retrospective design is acknowledged, but the discussion should more explicitly cover potential biases (e.g., incomplete anthropometric data, reliance on medical record accuracy, confounding by underlying undiagnosed conditions).

-     Exclusion criteria are well defined, but it would be helpful to clarify how “healthy” was determined—particularly whether all metabolic/hormonal disorders were actively ruled out or assumed absent unless clinically apparent.

7.     Statistical Analysis

-     Logistic regression appears appropriate; however, it is not clear whether potential collinearity between BMI and BMI-SDS was formally assessed prior to including both variables in the model.

-     Additional methodological details are needed—specifically, how missing BMI data were managed and whether any sensitivity analyses were performed. Providing this information would enhance transparency.

-     Finally, although numerous p-values are reported, no adjustment for multiple comparisons is mentioned. The rationale for the chosen approach should be clearly justified.

8.     Discussion

-     No continuation of the line numbering.

-     The Discussion could better differentiate between association and causality. For example, it is unclear whether low BMI is a causal risk factor for hypoglycaemia or merely a proxy for age-related physiology.

-     The finding that diarrheas' was associated with increased hypoglycaemia risk is intriguing but underexplored—possible mechanisms (fluid/electrolyte shifts, increased energy expenditure) should be discussed.

9.     Figures and Tables

-     Figures 2 and 3 are not visible and have no title in the main text.

-     Some legends need clarification (e.g., specify “CI” in figure legends for non-specialist readers).

Reviewers' comments:

Reviewer's Responses to Questions

**Comments to the Author**

1. Is the manuscript technically sound, and do the data support the conclusions?

Reviewer #1: Yes

2. Has the statistical analysis been performed appropriately and rigorously?

Reviewer #1: Yes

3. Have the authors made all data underlying the findings in their manuscript fully available?

Reviewer #1: No

4. Is the manuscript presented in an intelligible fashion and written in standard English?

Reviewer #1: Yes

Reviewer #1: This study is a retrospective single centre study addressing an important and clinically relevant question in children. Below are a few suggestions for the authors:

1) In the abstract it should be specified that this is a retrospective and single centre study.

2) In the abstract the result section could include ORs for the main preditors (BMI, diarrhoea and dehydration)

3) Line 41-42 could better read as (to avoid causal tone): Low BMI, low BMI-SDS and diarrhoea were associated with increased odds of hypoglycaemia.

4) In the introduction, could the authors differentiate between IKH and hypoglycaemia secondary to acute illness, they are overlapping but distinct entities.

5) Could the authors clarify why BMI was hypothesized as a potential risk factor? (what is the underlying pathophysiological reason)

6) In the methods section, could you elaborate further on the description of the forward stepwise selection and transformation of non-normal data? Were covariates checked for collinearity?

7) The BMI reference source used (Czech norms) may not represent Slovak chidren, could the authors justify the use of this reference?

8) For ethics consideration, please clarify that the informed consent was waived due to retrospective anonymized data.

9) For result section: the authors could split table 1 into 2 separate tables, anthropometric vs. biochemical to make it less overloaded

10) For the regression models, could the authors include odds ratios (ORs) with 95% CI ?

11) In the discussion, the speculation of low BMI being associated to smaller liver/muscle should be rephrased cautiously or deleted to avoid implying causality

12) Could the authors clarify the finding that diarrhoea is protective? Could this reflect late presentation, different dehydration mechanisms or measurement bias?

13) Regarding the age effects, although BMI correlated with age, the lack of an independent age effect in regression needs clarification; is there possibility of collinearity?

14) In the limitations: authors should discuss a) selection bias (how referral admission bias may inflate hypoglycaemia prevalence?), b) missing data (extent of missing data and how these were handled).

15) the section "Implications for clinical practice" currently reads as though the identified factors are definite predictors of hypoglycaemia, implying predictive certainty in clinical setting. However the study is a retrospective observational so it can only establish associations not causal relationships. So your findings are not yet validated as screening criteria or diagnostic predictors. So you could rephrase for example like this: Low BMI and abscence of diarrhoea were associated with increased odds of hypoglycaemia. These factors, readily indentifiable at presentation, may support early glucose screening in children with vomiting and dehydration. However, prospective validation is required before their use as predictive tools.

**Do you want your identity to be public for this peer review?** For information about this choice, including consent withdrawal, please see our Privacy Policy

Reviewer #1: No

---

## [Author Response · Author response to Decision Letter 1]

29 Dec 2025

Subject: Submission of Revised Manuscript (PONE-D-25-15589)

Title: Body mass index is associated with hypoglycaemia in children with acute vomiting and dehydration. (The title has been revised to better reflect the observational nature of the study and to avoid causal interpretation, in line with the editor’s and reviewer’s recommendations.)

Dear Editors,

We would like to sincerely thank you and the reviewer for the thorough, thoughtful, and constructive evaluation of our manuscript. The comments and suggestions provided were highly valuable and have substantially improved the clarity, methodological rigor, and clinical relevance of our work.

Below, we provide detailed, point-by-point responses to all comments raised by the academic editor and reviewer. All corresponding revisions in the manuscript are clearly indicated using track changes.

Kind regards,

Juraj Staník

on behalf of all co-authors

Response to the Editor – Journal Requirements

1. PLOS ONE style and formatting requirements

We will ensure that the revised manuscript fully complies with PLOS ONE formatting and style requirements, including file naming, in accordance with the official PLOS ONE templates provided by the journal.

All references have been carefully checked for consistency and accuracy, and a complete reference list has been added for Supplementary Table S1.

2. Participant consent and ethical considerations

All steps of the study were approved by the Ethics Committee of the National Institute of Children’s Diseases, Bratislava, Slovakia, and were conducted in accordance with the Declaration of Helsinki. The study was based on a retrospective analysis of routinely collected clinical data, which were partially anonymized prior to analysis.

In response to editorial and reviewer requests for clarification, the Ethics Committee issued a formal statement on December 10, 2025, confirming that the requirement for informed consent (written or verbal) from parents or legal guardians was waived due to the retrospective nature of the study and the use of partially anonymized data. The Ethics Committee also explicitly approved publication of the study results.

3. Full ethics statement in the Methods section

The following ethics statement has been included in the Methods section of the revised manuscript:

This study was approved by the Ethics Committee of the National Institute of Children’s Diseases, Bratislava, Slovakia, and adhered to the tenets outlined in the Declaration of Helsinki. The study was a retrospective analysis of routinely collected clinical data. The requirement for informed consent (written or verbal) from parents or legal guardians of participating children was waived by the Ethics Committee due to the retrospective design and the use of partially anonymized/de-identified data. Data were accessed for research on December 13, 2023. In response to editorial and reviewer queries, the Ethics Committee issued a formal clarification on December 10, 2025, confirming the waiver of informed consent and approving publication of the study results.

4. Data sharing restrictions and justification

The dataset underlying this study contains potentially identifiable clinical information, including exact age and sex, which precludes full anonymization to a level suitable for unrestricted public sharing. Public data sharing has therefore been restricted by the Ethics Committee of the National Institute of Children’s Diseases, Bratislava, Slovakia.

5. Data Availability Statement

The following Data Availability Statement has been included in the submission form:

The dataset contains potentially identifiable clinical information (including exact age and sex) and therefore cannot be fully anonymized to a standard suitable for unrestricted public sharing. The Ethics Committee of the National Institute of Children’s Diseases (Bratislava, Slovakia) has restricted public data sharing. De-identified data supporting the findings of this study are available upon reasonable request from the secretary of the Ethics Committee of the National Institute of Children’s Diseases, Bratislava, Slovakia (email: detska.klinika@nudch.eu), for researchers who meet the criteria for access to confidential data, subject to Ethics Committee approval and a data-sharing agreement where applicable.

6. Reviewer recommendations regarding additional citations

No reviewer explicitly recommended citation of specific additional publications beyond those already considered and appropriately cited in the revised manuscript.

Response to the Academic Editor

Additional Editor Comments

Summary Report

This retrospective study investigates risk factors for hypoglycemia in otherwise healthy children admitted with acute vomiting and dehydration. Among 560 patients aged 29 days to 17.9 years, 30.5% developed hypoglycemia (≤3.3 mmol/L), predominantly in the 2–7 year age range. Besides known contributors such as decreased oral intake and dehydration, the study identifies low BMI, low BMI-SDS, and diarrhoea as independent predictors. The findings offer a plausible mechanistic link between physiologically low BMI in early childhood and heightened susceptibility to hypoglycemia.

Response:

We sincerely thank the Academic Editor for this positive and thoughtful appraisal of our work. We very much appreciate the careful reading of the manuscript and the constructive suggestions, which substantially helped us improve the clarity, methodological transparency, and interpretative nuance of the revised version. Below, we provide a point-by-point response to each comment and outline the corresponding revisions made to the manuscript.

1. Language and readability

Comment:

The manuscript is generally clear, but occasional grammatical issues should be corrected (e.g., “IKT” should be “IKH”; “polystigmatisation” may be unclear in English and should be defined or replaced). Some sentences are overly long; breaking them into shorter units would improve readability.

Response:

We thank the editor for this observation. All typographical errors and unclear terms have been corrected, including replacement or removal of expressions that may be ambiguous in English. In addition, we have revised several long sentences throughout the manuscript and divided them into shorter, clearer units to improve readability.

2. Consistent terminology

Comment:

Ensure consistent use of terms (e.g., “hypoglycaemia ≤ 3.3 mmol/L” vs. “hypoglycaemia < 3.3 mmol/L”).

Response:

We appreciate this comment. The manuscript has been carefully reviewed, and terminology has been standardised throughout to ensure consistent use of definitions and thresholds for hypoglycaemia.

3. Definition of abbreviations

Comment:

Define all abbreviations at first use (e.g., AG for anion gap, SDS, IKH).

Response:

Thank you for this reminder. All abbreviations are now defined at their first occurrence, including in the Abstract, in accordance with journal guidelines.

4. Originality and significance

Comment:

The link between BMI and hypoglycemia is interesting and clinically relevant, especially since the 2–7 year age window coincides with physiologic BMI nadir. However, the claim of novelty could be better contextualized with more detailed comparison to prior studies.

Response:

We are very grateful to the editor for highlighting this important point. Both the Introduction and Discussion have been expanded to more explicitly contextualise our findings within the existing literature. We now clearly describe what has been reported in previous studies and specify how our work extends current knowledge, particularly by identifying BMI as an independent factor in a large cohort of children with illness-related hypoglycaemia.

5. Description of the study centre

Comment:

There is a poor description of the National Institute for Children’s Diseases in Bratislava.

Response:

Thank you for pointing this out. We have added a concise description of the study centre in the Methods section, specifying that the National Institute for Children’s Diseases in Bratislava is the largest tertiary paediatric referral centre in Slovakia, providing specialised care for children with chronic and rare diseases.

6. Study design and limitations

Comment:

The retrospective design is acknowledged, but potential biases should be discussed more explicitly.

Response:

We appreciate this suggestion. The Discussion, and particularly the Limitations section, has been substantially revised to more explicitly address potential sources of bias, including incomplete anthropometric data, reliance on routine medical records, and the possibility of undiagnosed underlying conditions.

Comment:

Clarify how “healthy” participants were defined.

Response:

Thank you for this clarifying question. The Methods section has been revised to explicitly state that metabolic or hormonal disorders were not systematically screened for, but were assumed to be absent unless previously diagnosed or clinically apparent.

7. Statistical analysis

Comment:

It is unclear whether collinearity between BMI and BMI-SDS was formally assessed.

Response:

We thank the editor for this important methodological comment. Collinearity among independent variables was assessed prior to multivariable modelling. Initially, collinearity was evaluated in SPSS using linear regression diagnostics, as SPSS does not provide collinearity statistics for logistic regression models. In this analysis, no moderate collinearity between BMI and BMI-SDS was detected (VIF = 5.2).

Subsequently, as part of the sensitivity analyses performed in R, collinearity was re-evaluated directly within the logistic regression framework, where variance inflation factors indicated substantial collinearity between BMI and BMI-SDS (VIF = 10.6). However, BMI-SDS was excluded from the multivariable models regardless, as it did not meet the Bonferroni-adjusted significance threshold in the univariable analyses. Therefore, collinearity did not affect the final multivariable models or the interpretation of the results.

Comment:

Clarify handling of missing BMI data and sensitivity analyses.

Response:

We are very grateful for this important suggestion. The Methods and Results sections now clearly describe how missing BMI data were handled. In addition to the complete-case analysis, we performed two sensitivity analyses: (i) a model excluding BMI and (ii) a multiple imputation analysis using chained equations with predictive mean matching implemented in R.

Results of these analyses were consistent with the primary findings and are presented in the Supplementary Materials (Table S2). We also added results comparing children with missing BMI to those with BMI below and above the cohort median, showing that missing BMI clustered in children with a similar hypoglycaemia risk profile as those with low BMI. We thank the editor for this highly relevant comment, which substantially strengthened the manuscript.

Comment:

Clarify adjustment for multiple comparisons.

Response:

We agree that this required explicit clarification. The Methods section now clearly states that Bonferroni correction was applied to univariable analyses (n = 30), resulting in a significance threshold of p < 0.0017. Only variables meeting this criterion were entered into multivariable models. No further adjustment was applied in multivariable analyses, as these were hypothesis-driven and based on a limited number of preselected variables.

8. Discussion

Comment:

Line numbering was incomplete.

Response:

Thank you for noting this. Line numbering has been corrected throughout the manuscript, including the Discussion section.

Comment:

Differentiate more clearly between association and causality.

Response:

We are very grateful to the editor for this crucial point. We now explicitly emphasise throughout the Discussion that our retrospective observational design allows assessment of associations only and does not support causal inference. In particular, we clarify that low BMI may act as a proxy for age-related physiological vulnerability rather than a causal risk factor per se. We have carefully reviewed the entire manuscript to remove language implying causality and have adjusted the manuscript title accordingly.

Comment:

The finding regarding diarrhoea warrants further exploration.

Response:

We appreciate this insightful comment. The Discussion has been expanded to explore possible mechanisms underlying the observed inverse association between diarrhoea and hypoglycaemia, including illness trajectory, resumption of oral intake, fluid and electrolyte balance, and energy demands. We also emphasise that this finding should be interpreted cautiously and requires prospective confirmation.

9. Figures and tables

Comment:

Figures 2 and 3 were not clearly visible and lacked titles.

Response:

Thank you for this observation. The figures have been reformatted with a white background to improve contrast and readability, and titles and references in the main text have been corrected accordingly.

Comment:

Some figure legends require clarification.

Response:

We thank the editor for this comment. All figure legends have been revised to clarify abbreviations, including explicit definition of confidence intervals (CI).

Closing statement

We again sincerely thank the Academic Editor for the careful evaluation and highly constructive feedback. The suggested revisions have significantly improved the clarity, methodological rigour, and interpretative balance of our manuscript. We trust that the revised version now fully addresses all concerns raised.

Reviewers’ comments

Reviewer #1 – General remarks

Comment:

This study is a retrospective single centre study addressing an important and clinically relevant question in children.

Response:

We sincerely thank the reviewer for the encouraging overall assessment and for the constructive, detailed suggestions. We greatly appreciate the emphasis on clinical relevance and careful interpretation of observational findings. Below, we address each comment point-by-point and describe the corresponding revisions made to the manuscript.

1) Abstract: specify retrospective single-centre design

Comment:

In the abstract it should be specified that this is a retrospective and single centre study.

Response:

Thank you. We have revised the Abstract to explicitly state that this was a retrospective, single-centre study.

2) Abstract: include ORs for main predictors

Comment:

In the abstract the result section could include ORs for the main predictors (BMI, diarrhoea and dehydration).

Response:

Thank you for this helpful suggestion. We have added odds ratios with 95% confidence intervals for the main predictors (BMI, diarrhoea, dehydration, and oral intake) to the Results section of the Abstract.

3) Avoid causal tone in Abstract

Comment:

Line 41-42 could better read as (to avoid causal tone): Low BMI, low BMI-SDS and diarrhoea were associated with increased odds of hypoglycaemia.

Response:

We agree and thank the reviewer for the proposed wording. We revised the sentence to use non-causal, association-based language, consistent with the retrospective observational design.

4) Differentiate IKH vs hypoglycaemia secondary to acute illness

Comment:

In the introduction, could the authors differentiate between IKH and hypoglycaemia secondary to acute illness, they are overlapping but distinct entities.

Response:

We are very grateful for this key comment, which significantly improved the conceptual clarity of the manuscript. We revised the Introduction to clearly distinguish:

(i) idiopathic ketotic hypoglycaemia (IKH) as a diagnosis typically established after recurrent episodes and a completed diagnostic evaluation excluding alternative causes, and

(ii) acute illness-related hypoglycaemia, which often represents a first documented episode occurring during vomiting/infection before a full diagnostic work-up is completed.

We also emphasise that these entities overlap in clinical practice, and that cohorts in acute illness settings may include children who later meet criteria f

---

## [Decision Letter · Decision Letter 1]

8 Jan 2026

Body mass index is associated with hypoglycaemia in children with acute vomiting and dehydration

PONE-D-25-15589R1

Dear Dr. Juraj Stanik

We’re pleased to inform you that your manuscript has been judged scientifically suitable for publication and will be formally accepted for publication once it meets all outstanding technical requirements.

Kind regards,

Tacilta Nhampossa

Academic Editor

PLOS One

Additional Editor Comments (optional):

Reviewers' comments:

Reviewer's Responses to Questions

**Comments to the Author**

Reviewer #1: All comments have been addressed

2. Is the manuscript technically sound, and do the data support the conclusions?

Reviewer #1: Yes

3. Has the statistical analysis been performed appropriately and rigorously?

Reviewer #1: Yes

4. Have the authors made all data underlying the findings in their manuscript fully available?

Reviewer #1: Yes

5. Is the manuscript presented in an intelligible fashion and written in standard English?

Reviewer #1: Yes

Reviewer #1: The authors have addressed all reviewers concerns and I believe the manuscript is now ready for publication.

**Do you want your identity to be public for this peer review?** For information about this choice, including consent withdrawal, please see our Privacy Policy

Reviewer #1: No

---

## [Editor Report · Acceptance letter]

PONE-D-25-15589R1

PLOS One

Dear Dr. Stanik,

I'm pleased to inform you that your manuscript has been deemed suitable for publication in PLOS One. Congratulations! Your manuscript is now being handed over to our production team.

Kind regards,

on behalf of

Dr. Tacilta Nhampossa

Academic Editor

PLOS One